# Learning Object Bounding Boxes for 3D Instance Segmentation on Point Clouds

**Bo Yang** [1]    **Jianan Wang** [2]    **Ronald Clark** [3]    **Qingyong Hu** [1]
**Sen Wang** [4]    **Andrew Markham** [1]    **Niki Trigoni** [1]

[1]University of Oxford    [2]DeepMind    [3]Imperial College London    [4]Heriot-Watt University
`firstname.lastname@cs.ox.ac.uk`

## Abstract

We propose a novel, conceptually simple and general framework for instance segmentation on 3D point clouds. Our method, called **3D-BoNet**, follows the simple design philosophy of per-point multilayer perceptrons (MLPs). The framework directly regresses 3D **bo**unding **bo**xes for all instances in a point cloud, while simultaneously predicting a point-level mask for each instance. It consists of a backbone network followed by two parallel network branches for 1) bounding box regression and 2) point mask prediction. 3D-BoNet is single-stage, anchor-free and end-to-end trainable. Moreover, it is remarkably computationally efficient as, unlike existing approaches, it does not require any post-processing steps such as non-maximum suppression, feature sampling, clustering or voting. Extensive experiments show that our approach surpasses existing work on both ScanNet and S3DIS datasets while being approximately $10\times$ more computationally efficient. Comprehensive ablation studies demonstrate the effectiveness of our design.

## 1    Introduction

Enabling machines to understand 3D scenes is a fundamental necessity for autonomous driving, augmented reality and robotics. Core problems on 3D geometric data such as point clouds include semantic segmentation, object detection and instance segmentation. Of these problems, instance segmentation has only started to be tackled in the literature. The primary obstacle is that point clouds are inherently unordered, unstructured and non-uniform. Widely used convolutional neural networks require the 3D point clouds to be voxelized, incurring high computational and memory costs.

The first neural algorithm to directly tackle 3D instance segmentation is SGPN [51], which learns to group per-point features through a similarity matrix. Similarly, ASIS [52], JSIS3D [35], MASC [31], 3D-BEVIS [8] and [29] apply the same per-point feature grouping pipeline to segment 3D instances. Mo *et al.* formulate the instance segmentation as a per-point feature classification problem in PartNet [33]. However, the learnt segments of these proposal-free methods do not have high objectness as they do not explicitly detect the object boundaries. In addition, they inevitably require a post-processing step such as mean-shift clustering [6] to obtain the final instance labels, which is computationally heavy. Another pipeline is the proposal-based 3D-SIS [15] and GSPN [59], which usually rely on two-stage training and the expensive non-maximum suppression to prune dense object proposals.

In this paper, we present an elegant, efficient and novel framework for 3D instance segmentation, where objects are loosely but uniquely detected through a single-forward stage using efficient MLPs, and then each instance is precisely segmented through a simple point-level binary classifier. To this end, we introduce a new bounding box prediction module together with a series of carefully designed loss functions to directly learn object boundaries. Our framework is significantly different from the existing proposal-based and proposal-free approaches, since we are able to efficiently segment all instances with high objectness, but without relying on expensive and dense object proposals. Our code and data are available at *https://github.com/Yang7879/3D-BoNet*.

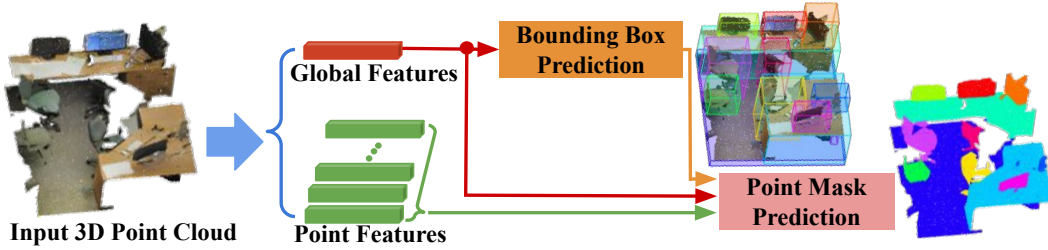

**Global Features**

**Bounding Box Prediction**

**Point Mask Prediction**

**Input 3D Point Cloud**

**Point Features**

Figure 1: The 3D-BoNet framework for instance segmentation on 3D point clouds.

As shown in Figure 1, our framework, called **3D-BoNet**, is a single-stage, anchor-free and end-to-end trainable neural architecture. It first uses an existing backbone network to extract a local feature vector for each point and a global feature vector for the whole input point cloud. The backbone is followed by two branches: 1) instance-level bounding box prediction, and 2) point-level mask prediction for instance segmentation.

The **bounding box prediction branch** is the core of our framework. This branch aims to predict a unique, unoriented and rectangular bounding box for each instance in a single forward stage, without relying on predefined spatial anchors or a region proposal network [40]. As shown in Figure 2, we believe that roughly drawing a 3D bounding box for an instance is relatively achievable, because the input point clouds explicitly include 3D geometry information, while it is extremely beneficial before tackling point-level instance segmentation since reasonable bounding boxes can guarantee high objectness for learnt segments. However, to learn instance boxes involves critical issues: 1) the number of total instances is variable,

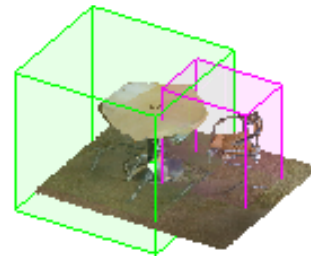

Figure 2: Rough instance boxes.

*i.e.*, from 1 to many, 2) there is no fixed order for all instances. These issues pose great challenges for correctly optimizing the network, because there is no information to directly link predicted boxes with ground truth labels to supervise the network. However, we show how to elegantly solve these issues. This box prediction branch simply takes the global feature vector as input and directly outputs a large and fixed number of bounding boxes together with confidence scores. These scores are used to indicate whether the box contains a valid instance or not. To supervise the network, we design a novel *bounding box association layer* followed by a *multi-criteria loss function*. Given a set of ground-truth instances, we need to determine which of the predicted boxes best fit them. We formulate this association process as an optimal assignment problem with an existing solver. After the boxes have been optimally associated, our multi-criteria loss function not only minimizes the Euclidean distance of paired boxes, but also maximizes the coverage of valid points inside of predicted boxes.

The predicted boxes together with point and global features are then fed into the subsequent **point mask prediction branch**, in order to predict a point-level binary mask for each instance. The purpose of this branch is to classify whether each point inside of a bounding box belongs to the valid instance or the background. Assuming the estimated instance box is reasonably good, it is very likely to obtain an accurate point mask, because this branch is simply to reject points that are not part of the detected instance. A random guess may bring about $50\%$ corrections.

Overall, our framework distinguishes from all existing 3D instance segmentation approaches in three folds. 1) Compared with the proposal-free pipeline, our method segments instance with high objectness by explicitly learning 3D object boundaries. 2) Compared with the widely-used proposal-based approaches, our framework does not require expensive and dense proposals. 3) Our framework is remarkably efficient, since the instance-level masks are learnt in a single-forward pass without requiring any post-processing steps. Our key contributions are:

- We propose a new framework for instance segmentation on 3D point clouds. The framework is single-stage, anchor-free and end-to-end trainable, without requiring any post-processing steps.

- We design a novel bounding box association layer followed by a multi-criteria loss function to supervise the box prediction branch.

- We demonstrate significant improvement over baselines and provide intuition behind our design choices through extensive ablation studies.

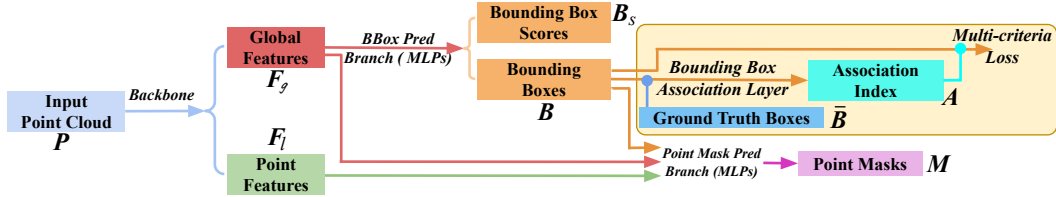

Figure 3: The general workflow of 3D-BoNet framework.

## 2 3D-BoNet

### 2.1 Overview

As shown in Figure 3, our framework consists of two branches on top of the backbone network. Given an input point cloud $P$ with $N$ points in total, $i.e.$, $P \in \mathbb{R}^{N \times k_0}$, where $k_0$ is the number of channels such as the location $\{x, y, z\}$ and color $\{r, g, b\}$ of each point, the **backbone network** extracts point local features, denoted as $F_l \in \mathbb{R}^{N \times k}$, and aggregates a global point cloud feature vector, denoted as $F_g \in \mathbb{R}^{1 \times k}$, where $k$ is the length of feature vectors.

The **bounding box prediction branch** simply takes the global feature vector $F_g$ as input, and directly regresses a predefined and fixed set of bounding boxes, denoted as $B$, and the corresponding box scores, denoted as $B_s$. We use ground truth bounding box information to supervise this branch. During training, the predicted bounding boxes $B$ and the ground truth boxes are fed into a *box association layer*. This layer aims to automatically associate a unique and most similar predicted bounding box to each ground truth box. The output of the association layer is a list of association index $A$. The indices reorganize the predicted boxes, such that each ground truth box is paired with a unique predicted box for subsequent loss calculation. The predicted bounding box scores are also reordered accordingly before calculating loss. The reordered predicted bounding boxes are then fed into the *multi-criteria loss function*. Basically, this loss function aims to not only minimize the Euclidean distance between each ground truth box and the associated predicted box, but also maximize the coverage of valid points inside of each predicted box. Note that, both the bounding box association layer and multi-criteria loss function are only designed for network training. They are discarded during testing. Eventually, this branch is able to predict a correct bounding box together with a box score for each instance directly.

In order to predict point-level binary mask for each instance, every predicted box together with previous local and global features, $i.e.$, $F_l$ and $F_g$, are further fed into the **point mask prediction branch**. This network branch is shared by all instances of different categories, and therefore extremely light and compact. Such class-agnostic approach inherently allows general segmentation across unseen categories.

### 2.2 Bounding Box Prediction

**Bounding Box Encoding:** In existing object detection networks, a bounding box is usually represented by the center location and the length of three dimensions [3], or the corresponding residuals [61] together with orientations. Instead, we parameterize the rectangular bounding box by only two min-max vertices for simplicity:

$$\{[x_{min}\ y_{min}\ z_{min}], [x_{max}\ y_{max}\ z_{max}]\}$$

**Neural Layers:** As shown in Figure 4, the global feature vector $F_g$ is fed through two fully connected layers with Leaky ReLU as the non-linear activation function. Then it is followed by another two parallel fully connected layers. One layer outputs a $6H$ dimensional vector, which is then reshaped as an $H \times 2 \times 3$ tensor. $H$ is a predefined and fixed number of bounding boxes that the whole network are expected to predict in maximum. The other layer outputs an $H$ dimensional vector followed by $sigmoid$ function to represent the bounding box scores. The higher the score, the more likely that the predicted box contains an instance, thus the box being more valid.

**Bounding Box Association Layer:** Given the previously predicted $H$ bounding boxes, $i.e.$, $B \in \mathbb{R}^{H \times 2 \times 3}$, it is not straightforward to take use of the ground truth boxes, denoted as $\bar{B} \in \mathbb{R}^{T \times 2 \times 3}$, to supervise the network, because there are no predefined anchors to trace each predicted box back to a corresponding ground truth box in our framework. Besides, for each input point cloud $P$, the number

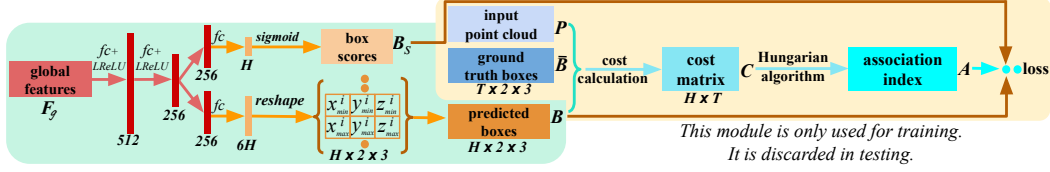

Figure 4: The architecture of bounding box regression branch. The predicted $H$ boxes are optimally associated with $T$ ground truth boxes before calculating the multi-criteria loss.

of ground truth boxes $T$ varies and it is usually different from the predefined number $H$, although we can safely assume the predefined number $H \geq T$ for all input point clouds. In addition, there is no box order for either predicted or ground truth boxes.

*Optimal Association Formulation:* To associate a unique predicted bounding box from $\boldsymbol{B}$ for each ground truth box of $\bar{\boldsymbol{B}}$, we formulate this association process as an optimal assignment problem. Formally, let $\boldsymbol{A}$ be a boolean association matrix where $\boldsymbol{A}_{i,j} = 1$ *iff* the $i^{th}$ predicted box is assigned to the $j^{th}$ ground truth box. $\boldsymbol{A}$ is also called association index in this paper. Let $\boldsymbol{C}$ be the association cost matrix where $\boldsymbol{C}_{i,j}$ represents the cost that the $i^{th}$ predicted box is assigned to the $j^{th}$ ground truth box. Basically, the cost $\boldsymbol{C}_{i,j}$ represents the similarity between two boxes; the less the cost, the more similar the two boxes. Therefore, the bounding box association problem is to find the optimal assignment matrix $\boldsymbol{A}$ with the minimal cost overall:

$$\boldsymbol{A} = \arg\min_{\boldsymbol{A}} \sum_{i=1}^{H} \sum_{j=1}^{T} \boldsymbol{C}_{i,j} \boldsymbol{A}_{i,j} \quad \text{subject to} \sum_{i=1}^{H} \boldsymbol{A}_{i,j} = 1, \sum_{j=1}^{T} \boldsymbol{A}_{i,j} \leq 1, j \in \{1..T\}, i \in \{1..H\} \quad (1)$$

To solve the above optimal association problem, the existing Hungarian algorithm [21; 22] is applied.

*Association Matrix Calculation:* To evaluate the similarity between the $i^{th}$ predicted box and the $j^{th}$ ground truth box, a simple and intuitive criterion is the Euclidean distance between two pairs of min-max vertices. However, it is not optimal. Basically, we want the predicted box to include as many valid points as possible. As illustrated in Figure 5, the input point cloud is usually sparse and distributed non-uniformly in 3D space. Regarding the same ground truth box #0 (blue), the candidate box #2 (red) is believed to be much better than the candidate #1 (black), because the box #2 has more valid points overlapped with #0. Therefore, the coverage of valid points should be included to calculate the cost matrix $\boldsymbol{C}$. In this paper, we consider the following three criteria:

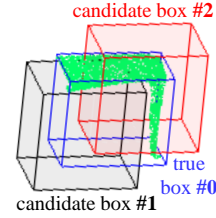

Figure 5: A sparse input point cloud.

(1) Euclidean Distance between Vertices. Formally, the cost between the $i^{th}$ predicted box $\boldsymbol{B}_i$ and the $j^{th}$ ground truth box $\bar{\boldsymbol{B}}_j$ is calculated as follows:

$$\boldsymbol{C}_{i,j}^{ed} = \frac{1}{6} \sum (\boldsymbol{B}_i - \bar{\boldsymbol{B}}_j)^2 \quad (2)$$

(2) Soft Intersection-over-Union on Points. Given the input point cloud $\boldsymbol{P}$ and the $j^{th}$ ground truth instance box $\bar{\boldsymbol{B}}_j$, it is able to directly obtain a hard-binary vector $\bar{\boldsymbol{q}}_j \in \mathbb{R}^N$ to represent whether each point is inside of the box or not, where '1' indicates the point being inside and '0' outside. However, for a specific $i^{th}$ predicted box of the same input point cloud $\boldsymbol{P}$, to directly obtain a similar hard-binary vector would result in the framework being non-differentiable, due to the discretization operation. Therefore, we

**Algorithm 1** An algorithm to calculate point-in-pred-box-probability. $H$ is the number of predicted bounding boxes $\boldsymbol{B}$, $N$ is the number of points in point cloud $\boldsymbol{P}$, $\theta_1$ and $\theta_2$ are hyperparameters for numerical stability. We use $\theta_1 = 100$, $\theta_2 = 20$ in all our implementation.

---

**for** $i \leftarrow 1$ to $H$ **do**
  - the $i^{th}$ box min-vertex $\boldsymbol{B}_{min}^i = [x_{min}^i \ y_{min}^i \ z_{min}^i]$.
  - the $i^{th}$ box max-vertex $\boldsymbol{B}_{max}^i = [x_{max}^i \ y_{max}^i \ z_{max}^i]$.

  **for** $n \leftarrow 1$ to $N$ **do**
    - the $n^{th}$ point location $\boldsymbol{P}^n = [x^n \ y^n \ z^n]$.
    - step 1: $\boldsymbol{\Delta}_{xyz} \leftarrow (\boldsymbol{B}_{min}^i - \boldsymbol{P}^n)(\boldsymbol{P}^n - \boldsymbol{B}_{max}^i)$.
    - step 2: $\boldsymbol{\Delta}_{xyz} \leftarrow max\left[min(\theta_1 \boldsymbol{\Delta}_{xyz}, \theta_2), -\theta_2\right]$.
    - step 3: probability $\boldsymbol{p}_{xyz} = \frac{1}{1+\exp(-\boldsymbol{\Delta}_{xyz})}$.
    - step 4: point probability $q_i^n = min(\boldsymbol{p}_{xyz})$.
  - obtain the soft-binary vector $\boldsymbol{q}_i = [q_i^1 \cdots q_i^N]$.

The above two loops are only for illustration. They are easily replaced by standard and efficient matrix operations.

---

introduce a differentiable yet simple algorithm 1 to obtain a similar but soft-binary vector $\boldsymbol{q}_i$, called **point-in-pred-box-probability**, where all values are in the range $(0, 1)$. The deeper the corresponding point is inside of the box, the higher the value. The farther away the point is outside, the smaller

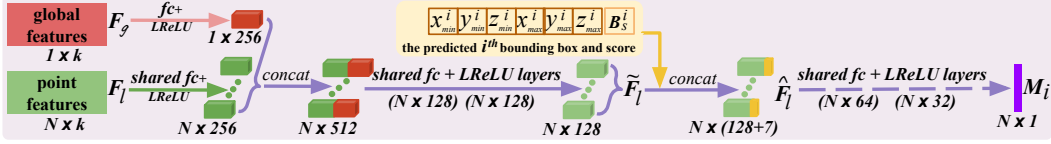

Figure 6: The architecture of point mask prediction branch. The point features are fused with each bounding box and score, after which a point-level binary mask is predicted for each instance.

the value. Formally, the Soft Intersection-over-Union (sIoU) cost between the $i^{th}$ predicted box and the $j^{th}$ ground truth box is defined as follows:

$$C_{i,j}^{sIoU} = \frac{-\sum_{n=1}^{N}(q_i^n * \bar{q}_j^n)}{\sum_{n=1}^{N} q_i^n + \sum_{n=1}^{N} \bar{q}_j^n - \sum_{n=1}^{N}(q_i^n * \bar{q}_j^n)} \tag{3}$$

where $q_i^n$ and $\bar{q}_j^n$ are the $n^{th}$ values of $\boldsymbol{q}_i$ and $\bar{\boldsymbol{q}}_j$.

(3) Cross-Entropy Score. In addition, we also consider the cross-entropy score between $\boldsymbol{q}_i$ and $\bar{\boldsymbol{q}}_j$. Being different from sIoU cost which prefers tighter boxes, this score represents how confident a predicted bounding box is able to include valid points as many as possible. It prefers larger and more inclusive boxes, and is formally defined as:

$$C_{i,j}^{ces} = -\frac{1}{N} \sum_{n=1}^{N} \left[ \bar{q}_j^n \log q_i^n + (1 - \bar{q}_j^n) \log(1 - q_i^n) \right] \tag{4}$$

Overall, the criterion (1) guarantees the geometric boundaries for learnt boxes and criteria (2)(3) maximize the coverage of valid points and overcome the non-uniformity as illustrated in Figure 5. The final association cost between the $i^{th}$ predicted box and the $j^{th}$ ground truth box is defined as:

$$C_{i,j} = C_{i,j}^{ed} + C_{i,j}^{sIoU} + C_{i,j}^{ces} \tag{5}$$

**Loss Functions** After the bounding box association layer, both the predicted boxes $\boldsymbol{B}$ and scores $\boldsymbol{B}_s$ are reordered using the association index $\boldsymbol{A}$, such that the first predicted $T$ boxes and scores are well paired with the $T$ ground truth boxes.

*Multi-criteria Loss for Box Prediction*: The previous association layer finds the most similar predicted box for each ground truth box according to the minimal cost including: 1) vertex Euclidean distance, 2) sIoU cost on points, and 3) cross-entropy score. Therefore, the loss function for bounding box prediction is naturally designed to consistently minimize those cost. It is formally defined as follows:

$$\ell_{bbox} = \frac{1}{T} \sum_{t=1}^{T} (C_{t,t}^{ed} + C_{t,t}^{sIoU} + C_{t,t}^{ces}) \tag{6}$$

where $C_{t,t}^{ed}$, $C_{t,t}^{sIoU}$ and $C_{t,t}^{ces}$ are the cost of $t^{th}$ paired boxes. Note that, we only minimize the cost of $T$ paired boxes; the remaining $H - T$ predicted boxes are ignored because there is no corresponding ground truth for them. Therefore, this box prediction sub-branch is agnostic to the predefined value of $H$. Here raises an issue. Since the $H - T$ negative predictions are not penalized, it might be possible that the network predicts multiple similar boxes for a single instance. Fortunately, the loss function for the parallel box score prediction is able to alleviate this problem.

*Loss for Box Score Prediction*: The predicted box scores aim to indicate the validity of the corresponding predicted boxes. After being reordered by the association index $\boldsymbol{A}$, the ground truth scores for the first $T$ scores are all '1', and '0' for the remaining invalid $H - T$ scores. We use cross-entropy loss for this binary classification task:

$$\ell_{bbs} = -\frac{1}{H} \left[ \sum_{t=1}^{T} \log \boldsymbol{B}_s^t + \sum_{t=T+1}^{H} \log(1 - \boldsymbol{B}_s^t) \right] \tag{7}$$

where $\boldsymbol{B}_s^t$ is the $t^{th}$ predicted score after being associated. Basically, this loss function rewards the correctly predicted bounding boxes, while implicitly penalizing the cases where multiple similar boxes are regressed for a single instance.

## 2.3 Point Mask Prediction

Given the predicted bounding boxes $\boldsymbol{B}$, the learnt point features $\boldsymbol{F}_l$ and global features $\boldsymbol{F}_g$, the point mask prediction branch processes each bounding box individually with shared neural layers.

Table 1: Instance segmentation results on ScanNet(v2) benchmark (hidden test set). The metric is AP(%) with IoU threshold 0.5. Accessed on 2 June 2019.

| | mean | bathtub | bed | bookshelf | cabinet | chair | counter | curtain | desk | door | other | picture | refrig | showerCur | sink | sofa | table | toilet | window |
|---|---|---|---|---|---|---|---|---|---|---|---|---|---|---|---|---|---|---|---|
| MaskRCNN [13] | 5.8 | 33.3 | 0.2 | 0.0 | 5.3 | 0.2 | 0.2 | 2.1 | 0.0 | 4.5 | 2.4 | 23.8 | 6.5 | 0.0 | 1.4 | 10.7 | 2.0 | 11.0 | 0.6 |
| SGPN [51] | 14.3 | 20.8 | 39.0 | 16.9 | 6.5 | 27.5 | 2.9 | 6.9 | 0.0 | 8.7 | 4.3 | 1.4 | 2.7 | 0.0 | 11.2 | 35.1 | 16.8 | 43.8 | 13.8 |
| 3D-BEVIS [8] | 24.8 | 66.7 | 56.6 | 7.6 | 3.5 | 39.4 | 2.7 | 3.5 | 9.8 | 9.9 | 3.0 | 2.5 | 9.8 | 37.5 | 12.6 | 60.4 | 18.1 | 85.4 | 17.1 |
| R-PointNet [59] | 30.6 | 50.0 | 40.5 | 31.1 | 34.8 | 58.9 | 5.4 | 6.8 | 12.6 | 28.3 | 29.0 | 2.8 | 21.9 | 21.4 | 33.1 | 39.6 | 27.5 | 82.1 | 24.5 |
| UNet-Backbone [29] | 31.9 | 66.7 | 71.5 | 23.3 | 18.9 | 47.9 | 0.8 | 21.8 | 6.7 | 20.1 | 17.3 | 10.7 | 12.3 | 43.8 | 15.0 | 61.5 | 35.5 | 91.6 | 9.3 |
| 3D-SIS (5 views) [15] | 38.2 | **100.0** | 43.2 | 24.5 | 19.0 | 57.7 | 1.3 | 26.3 | 3.3 | 32.0 | 24.0 | 7.5 | 42.2 | 85.7 | 11.7 | **69.9** | 27.1 | 88.3 | 23.5 |
| MASC [31] | 44.7 | 52.8 | 55.5 | 38.1 | **38.2** | 63.3 | 0.2 | 50.9 | 26.0 | 36.1 | 43.2 | 32.7 | **45.1** | 57.1 | 36.7 | 63.9 | 38.6 | **98.0** | 27.6 |
| ResNet-Backbone [29] | 45.9 | **100.0** | **73.7** | 15.9 | 25.9 | 58.7 | **13.8** | 47.5 | 21.7 | **41.6** | 40.8 | 12.8 | 31.5 | 71.4 | 41.1 | 53.6 | **59.0** | 87.3 | 30.4 |
| PanopticFusion [34] | 47.8 | 66.7 | 71.2 | **59.5** | 25.9 | 55.0 | 0.0 | 61.3 | 17.5 | 25.0 | **43.4** | **43.7** | 41.1 | 85.7 | **48.5** | 59.1 | 26.7 | 94.4 | 35.9 |
| MTML | 48.1 | **100.0** | 66.6 | 37.7 | 27.2 | **70.9** | 0.1 | 57.9 | 25.4 | 36.1 | 31.8 | 9.5 | 43.2 | **100.0** | 18.4 | 60.1 | 48.7 | 93.8 | 38.4 |
| **3D-BoNet(Ours)** | **48.8** | **100.0** | 67.2 | 59.0 | 30.1 | 48.4 | 9.8 | **62.0** | **30.6** | 34.1 | 25.9 | 12.5 | 43.4 | 79.6 | 40.2 | 49.9 | 51.3 | 90.9 | **43.9** |

**Neural Layers:** As shown in Figure 6, both the point and global features are compressed to be 256 dimensional vectors through fully connected layers, before being concatenated and further compressed to be 128 dimensional mixed point features $\widetilde{F}_l$. For the $i^{th}$ predicted bounding box $B_i$, the estimated vertices and score are fused with features $\widetilde{F}_l$ through concatenation, producing box-aware features $\widehat{F}_l$. These features are then fed through shared layers, predicting a point-level binary mask, denoted as $M_i$. We use $sigmoid$ as the last activation function. This simple box fusing approach is extremely computationally efficient, compared with the commonly used RoIAlign in prior art [59; 15; 13] which involves the expensive point feature sampling and alignment.

**Loss Function:** The predicted instance masks $M$ are similarly associated with the ground truth masks according to the previous association index $A$. Due to the imbalance of instance and background point numbers, we use focal loss [30] with default hyper-parameters instead of the standard cross-entropy loss to optimize this branch. Only the valid $T$ paired masks are used for the loss $\ell_{pmask}$.

## 2.4 End-to-End Implementation

While our framework is not restricted to any point cloud network, we adopt PointNet++ [39] as the backbone to learn the local and global features. Parallelly, another separate branch is implemented to learn per-point semantics with the standard $softmax$ cross-entropy loss function $\ell_{sem}$. The architecture of the backbone and semantic branch is the same as used in [51]. Given an input point cloud $P$, the above three branches are linked and end-to-end trained using a single combined multi-task loss:

$$\ell_{all} = \ell_{sem} + \ell_{bbox} + \ell_{bbs} + \ell_{pmask} \tag{8}$$

We use Adam solver [19] with its default hyper-parameters for optimization. Initial learning rate is set to $5e^{-4}$ and then divided by 2 every 20 epochs. The whole network is trained on a Titan X GPU from scratch. We use the same settings for all experiments, which guarantees the reproducibility of our framework.

## 3 Experiments

### 3.1 Evaluation on ScanNet Benchmark

We first evaluate our approach on ScanNet(v2) 3D semantic instance segmentation benchmark [7]. Similar to SGPN [51], we divide the raw input point clouds into $1m \times 1m$ blocks for training, while using all points for testing followed by the BlockMerging algorithm [51] to assemble blocks into complete 3D scenes. In our experiment, we observe that the performance of the vanilla PointNet++ based semantic prediction sub-branch is limited and unable to provide satisfactory semantics. Thanks to the flexibility of our framework, we therefore easily train a parallel SCN network [11] to estimate more accurate per-point semantic labels for the predicted instances of our 3D-BoNet. The average precision (AP) with an IoU threshold 0.5 is used as the evaluation metric.

We compare with the leading approaches on 18 object categories in Table 1. Particularly, the SGPN [51], 3D-BEVIS [8], MASC [31] and [29] are point feature clustering based approaches; the R-PointNet [59] learns to generate dense object proposals followed by point-level segmentation; 3D-SIS [15] is a proposal-based approach using both point clouds and color images as input. PanopticFusion [34] learns to segment instances on multiple 2D images by Mask-RCNN [13] and then uses the SLAM system to reproject back to 3D space. Our approach surpasses them all using point clouds only. Remarkably, our framework performs relatively satisfactory on all categories without preferring specific classes, demonstrating the superiority of our framework.

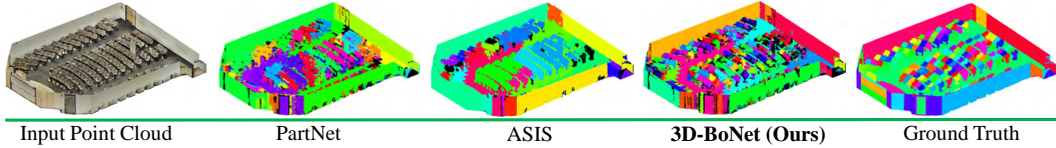

| Input Point Cloud | PartNet | ASIS | **3D-BoNet (Ours)** | Ground Truth |

Figure 7: This shows a lecture room with hundreds of objects (*e.g.*, chairs, tables), highlighting the challenge of instance segmentation. Different color indicates different instance. The same instance may not have the same color. Our framework predicts more precise instance labels than others.

## 3.2 Evaluation on S3DIS Dataset

We further evaluate the semantic instance segmentation of our framework on S3DIS [1], which consists of 3D complete scans from 271 rooms belonging to 6 large areas. Our data preprocessing and experimental settings strictly follow PointNet [38], SGPN [51], ASIS [52], and JSIS3D [35]. In our experiments, $H$ is set as 24 and we follow the 6-fold evaluation [1; 52].

Table 2: Instance segmentation results on S3DIS dataset.

|  | mPrec | mRec |
|---|---|---|
| PartNet [33] | 56.4 | 43.4 |
| ASIS [52] | 63.6 | 47.5 |
| **3D-BoNet (Ours)** | **65.6** | **47.6** |

We compare with ASIS [52], the state of art on S3DIS, and the PartNet baseline [33]. For fair comparison, we carefully train the PartNet baseline with the same PointNet++ backbone and other settings as used in our framework. For evaluation, the classical metrics mean precision (mPrec) and mean recall (mRec) with IoU threshold 0.5 are reported. Note that, we use the same BlockMerging algorithm [51] to merge the instances from different blocks for both our approach and the PartNet baseline. The final scores are averaged across the total 13 categories. Table 2 presents the mPrec/mRec scores and Figure 7 shows qualitative results. Our method surpasses PartNet baseline [33] by large margins, and also outperforms ASIS [52], but not significantly, mainly because our semantic prediction branch (vanilla PointNet++ based) is inferior to ASIS which tightly fuses semantic and instance features for mutual optimization. We leave the feature fusion as our future exploration.

## 3.3 Ablation Study

To evaluate the effectiveness of each component of our framework, we conduct 6 groups of ablation experiments on the largest Area 5 of S3DIS dataset.

(1) **Remove Box Score Prediction Sub-branch.** Basically, the box score serves as an indicator and regularizer for valid bounding box prediction. After removing it, we train the network with:

$$\ell_{ab1} = \ell_{sem} + \ell_{bbox} + \ell_{pmask}$$

Initially, the multi-criteria loss function is a simple unweighted combination of the Euclidean distance, the

Table 3: Instance segmentation results of all ablation experiments on Area 5 of S3DIS.

|  | mPrec | mRec |
|---|---|---|
| (1) Remove Box Score Sub-branch | 50.9 | 40.9 |
| (2) Euclidean Distance Only | 53.8 | **41.1** |
| (3) Soft IoU Cost Only | 55.2 | 40.6 |
| (4) Cross-Entropy Score Only | 51.8 | 37.8 |
| (5) Do Not Supervise Box Prediction | 37.3 | 28.5 |
| (6) Remove Focal Loss | 50.8 | 39.2 |
| **(7) The Full Framework** | **57.5** | 40.2 |

soft IoU cost, and the cross-entropy score. However, this may not be optimal, because the density of input point clouds is usually inconsistent and tends to prefer different criterion. We conduct the below 3 groups of experiments on ablated bounding box loss function.

(2)-(4) **Use Single Criterion.** Only one criterion is used for the box association and loss $\ell_{bbox}$.

$$\ell_{ab2} = \ell_{sem} + \frac{1}{T}\sum_{t=1}^{T} \boldsymbol{C}_{t,t}^{ed} + \ell_{bbs} + \ell_{pmask} \quad \cdots \quad \ell_{ab4} = \ell_{sem} + \frac{1}{T}\sum_{t=1}^{T} \boldsymbol{C}_{t,t}^{ces} + \ell_{bbs} + \ell_{pmask}$$

(5) **Do Not Supervise Box Prediction.** The predicted boxes are still associated according to the three criteria, but we remove the box supervision signal. The framework is trained with:

$$\ell_{ab5} = \ell_{sem} + \ell_{bbs} + \ell_{pmask}$$

(6) **Remove Focal Loss for Point Mask Prediction.** In the point mask prediction branch, the focal loss is replaced by the standard cross-entropy loss for comparison.

**Analysis.** Table 3 shows the scores for ablation experiments. (1) The box score sub-branch indeed benefits the overall instance segmentation performance, as it tends to penalize duplicated box predictions. (2) Compared with Euclidean distance and cross-entropy score, the sIoU cost tends to be better for box association and supervision, thanks to our differentiable Algorithm 1. As the three individual criteria prefer different types of point structures, a simple combination of three criteria

may not always be optimal on a specific dataset. (3) Without the supervision for box prediction, the performance drops significantly, primarily because the network is unable to infer satisfactory instance 3D boundaries and the quality of predicted point masks deteriorates accordingly. (4) Compared with focal loss, the standard cross entropy loss is less effective for point mask prediction due to the imbalance of instance and background point numbers.

## 3.4 Computation Analysis

(1) For point feature clustering based approaches including SGPN [51], ASIS [52], JSIS3D [35], 3D-BEVIS [8], MASC [31], and [29], the computation complexity of the post clustering algorithm such as Mean Shift [6] tends towards $\mathcal{O}(TN^2)$, where $T$ is the number of instances and $N$ is the number of input points. (2) For dense proposal-based methods including GSPN [59], 3D-SIS [15] and PanopticFusion [34], region proposal network and non-maximum suppression are usually required to generate and prune the dense proposals, which is computationally expensive [34]. (3) Both PartNet baseline [33] and our 3D-BoNet have similar efficient computation complexity $\mathcal{O}(N)$. Empirically, our 3D-BoNet takes around $20$ ms GPU time to process $4k$ points, while most approaches in (1)(2) need more than 200ms GPU/CPU time to process the same number of points.

## 4 Related Work

To extract features from 3D point clouds, traditional approaches usually craft features manually [5; 43]. Recent learning based approaches mainly include voxel-based [43; 47; 42; 24; 41; 11; 4] and point-based schemes [38; 20; 14; 17; 46].

**Semantic Segmentation** PointNet [38] shows leading results on classification and semantic segmentation, but it does not capture context features. To address it, a number of approaches [39; 58; 44; 32; 56; 50; 27; 18] have been proposed recently. Another pipeline is convolutional kernel based approaches [56; 28; 48]. Basically, most of these approaches can be used as our backbone network, and parallelly trained with our 3D-BoNet to learn per-point semantics.

**Object Detection** The common way to detect objects in 3D point clouds is to project points onto 2D images to regress bounding boxes [26; 49; 3; 57; 60; 54]. Detection performance is further improved by fusing RGB images in [3; 55; 37; 53]. Point clouds can be also divided into voxels for object detection [9; 25; 61]. However, most of these approaches rely on predefined anchors and the two-stage region proposal network [40]. It is inefficient to extend them on 3D point clouds. Without relying on anchors, the recent PointRCNN [45] learns to detect via foreground point segmentation, and the VoteNet [36] detects objects via point feature grouping, sampling and voting. By contrast, our box prediction branch is completely different from them all. Our framework directly regresses 3D object bounding boxes from the compact global features through a single forward pass.

**Instance Segmentation** SGPN [51] is the first neural algorithm to segment instances on 3D point clouds by grouping the point-level embeddings. ASIS [52], JSIS3D [35], MASC [31], 3D-BEVIS [8] and [29] use the same strategy to group point-level features for instance segmentation. Mo *et al*. introduce a segmentation algorithm in PartNet [33] by classifying point features. However, the learnt segments of these proposal-free methods do not have high objectness as it does not explicitly detect object boundaries. By drawing on the successful 2D RPN [40] and RoI [13], GSPN [59] and 3D-SIS [15] are proposal-based methods for 3D instance segmentation. However, they usually rely on two-stage training and a post-processing step for dense proposal pruning. By contrast, our framework directly predicts a point-level mask for each instance within an explicitly detected object boundary, without requiring any post-processing steps.

## 5 Conclusion

Our framework is simple, effective and efficient for instance segmentation on 3D point clouds. However, it also has some limitations which lead to the future work. (1) Instead of using unweighted combination of three criteria, it is better to design a module to automatically learn the weights, so to adapt to different types of input point clouds. (2) Instead of training a separate branch for semantic prediction, more advanced feature fusion modules can be introduced to mutually improve both semantic and instance segmentation. (3) Our framework follows the MLP design and is therefore agnostic to the number and order of input points. It is desirable to directly train and test on large-scale input point clouds instead of the divided small blocks, by drawing on the recent work [10][23][16].

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
