[Supplementary Material]

## *Appendix*

### A    Experiments on ScanNet Benchmark

ScanNet(v2) consists of 1613 complete 3D scenes acquired from real-world indoor spaces. The official split has 1201 training scenes, 312 validation scenes and 100 hidden testing scenes. The original large point clouds are divided into $1m \times 1m$ blocks with $0.5m$ overlapped between neighbouring blocks. This data proprocessing step is the same as being used by PointNet [38] for the S3DIS dataset. We sample 4096 points from each block for training, but use all points of a block for testing. Each point is represented by a 9D vector *(normalized xyz in the block, rgb, normalized xyz in the room)*. $H$ is set as 20 in our experiments. We train our 3D-BoNet to predict object bounding boxes and point-level masks, and parallelly train an officially released ResNet-based SCN network [11] to predict point-level semantic labels.

Figure 8 shows qualitative results of our 3D-BoNet for instance segmentation on ScanNet validation split. It can be seen that our approach tends to predict complete object instances, instead of inferring tiny and but invalid fragments. This demonstrates that our framework indeed guarantees high objectness for segmented instances. The red circles showcase the failure cases, where the very similar instances are unable to be well segmented by our approach.

**Input Point Clouds    Predicted Instance Labels    Ground Truth**

Figure 8: Qualitative results of our approach for instance segmentation on ScanNet(v2) validation split. Different color indicates different instance. The same instance may not be indicated by the same color. Black points are uninterested and belong to none of the 18 object categories.

## B    Experiments on S3DIS Dataset

The original large point clouds are divided into $1m \times 1m$ blocks with $0.5m$ overlapped between neighbouring blocks. It is the same as being used in PointNet [38]. We sample $4096$ points from each block for training, but use all points of a block for testing. Each point is represented by a 9D vector *(normalized xyz in the block, rgb, normalized xyz in the room)*. $H$ is set as $24$ in our experiments. We train our 3D-BoNet to predict object bounding boxes and point-level masks, and parallelly train a vanilla PointNet++ based sub-branch to predict point-level semantic labels. Particularly, all the semantic, bounding box and point mask sub-branches share the same PointNet++ backbone to extract point features, and are end-to-end trained from scratch.

Figure 9 shows the training curves of our proposed loss functions on Areas (1,2,3,4,6) of S3DIS dataset. It demonstrates that all the proposed loss functions are able to converge consistently, thus jointly optimizing the semantic segmentation, bounding box prediction, and point mask prediction branches in an end-to-end fashion.

Figure 10 presents the qualitative results of predicted bounding boxes and scores. It can be seen that the predicted boxes are not necessarily tight and precise. Instead, they tend to be inclusive but with high objectness. Fundamentally, this highlights the simple but effective concept of our bounding box prediction network. Given these bounded points, it is extremely easy to segment the instance inside.

Figure 11 visualizes the predicted instance masks, where the black points have $\sim 0$ probability and the brighter points have $\sim 1$ probability to be an instance within each predicted mask.

Figure 9: Training losses on S3DIS Areas (1,2,3,4,6).

Figure 10: Qualitative results of predicted bounding boxes and scores on S3DIS Area 2. The point clouds inside of the blue boxes are fed into our framework which then estimates the red boxes to roughly detect instances. The tight blue boxes are the ground truth.

| Input PC | *Pred Mask #1* | *Pred Mask #2* | *Pred Mask #3* | *Pred Mask #4* | GT Masks |

Figure 11: Qualitative results of predicted instance masks.

## C  Experiments for Computation Efficiency

Table 4 compares the time consumption of four existing approaches using their released codes on the validation split (312 scenes) of ScanNet(v2) dataset. SGPN [51], ASIS [52], GSPN [59] and our 3D-BoNet are implemented by Tensorflow 1.4, 3D-SIS [15] by Pytorch 0.4. All approaches are running on a single Titan X GPU and the pre/post-processing steps on an i7 CPU core with a single thread. Note that 3D-SIS automatically uses CPU for computing when some large scenes are unable to be processed by the single GPU. Overall, our approach is much more computationally efficient than existing methods, even achieving up to 20× faster than ASIS [52].

Table 4: Time consumption of different approaches on the validation split (312 scenes) of ScanNet(v2) (seconds).

|  | SGPN [51] | ASIS [52] | GSPN [59] | 3D-SIS [15] | **3D-BoNet(Ours)** |
|---|---|---|---|---|---|
|  | network(GPU): 650 | network(GPU): 650 | network(GPU): 500 | voxelization, projection, | network(GPU): 650 |
|  | group merging(CPU): 46562 | mean shift(CPU): 53886 | point sampling(GPU): 2995 | network, etc. (GPU+CPU): | *SCN (GPU parallel): 208* |
|  | block merging(CPU): 2221 | block merging(CPU): 2221 | neighbour search(CPU): 468 | 38841 | block merging(CPU): 2221 |
| total | 49433 | 56757 | 3963 | 38841 | **2871** |

## D  Gradient Estimation for Hungarian Algorithm

Given the predicted bounding boxes, $\mathbf{B}$, and ground-truth boxes, $\bar{\mathbf{B}}$, we compute the assignment cost matrix, $\mathbf{C}$. This matrix is converted to a permutation matrix, $\mathbf{A}$, using the Hungarian algorithm. Here we focus on the euclidean distance component of the loss, $\mathbf{C}^{ed}$. The derivative of our loss component w.r.t the network parameters, $\theta$, in matrix form is:

$$\frac{\partial \mathbf{C}^{ed}}{\partial \theta} = -2(\mathbf{AB} - \bar{\mathbf{B}}) \left[ \mathbf{A} + \frac{\partial \mathbf{A}}{\partial \mathbf{C}} \frac{\partial \mathbf{C}}{\partial \mathbf{B}} \mathbf{B} \right]^{T} \frac{\partial \mathbf{B}}{\partial \theta} \tag{9}$$

The components are easily computable except for $\frac{\partial \mathbf{A}}{\partial \mathbf{C}}$ which is the gradient of the permutation w.r.t the assignment cost matrix which is zero nearly everywhere. In our implementation, we found that the network is able to converge when setting this term to zero.

However, convergence could be sped up using the straight-through-estimator [2] , which assumes that the gradient of the rounding is identity (or a small constant), $\frac{\partial \mathbf{A}}{\partial \mathbf{C}} = \mathbb{1}$. This would speed up convergence as it allows both the error in the bounding box alignment (1st term of Eq. 9) to be backpropagated and the assignment to be reinforced (2nd term of Eq. 9). This approach has been shown to work well in practice for many problems including for differentiating through permutations for solving combinatorial optimization problems and for training binary neural networks . More complex approaches could also be used in our framework for computing the gradient of the assignment such as [12] which uses a Plackett-Luce distribution over permutations and a reparameterized gradient estimator.

# E    Generalization to Unseen Scenes and Categories

Our framework learns the object bounding boxes and point masks from raw point clouds without coupling with semantic information, which inherently allows the generalization across new categories and scenes. We conduct extra experiments to qualitatively demonstrate the generality of our framework. In particular, we use the well-trained model from S3DIS dataset (Areas 1/2/3/4/6) to directly test on the validation split of ScanNet(v2) dataset. Since ScanNet dataset consists of much more object categories than S3DIS dataset, there are a number of categories (*e.g.*, toilet, desk, sink, bathtub) that the trained model has never seen before.

As shown in Figure 12, our model is still able to predict high-quality instance labels even though the scenes and some object categories have not been seen before. This shows that our model does not simply fit the training dataset. Instead, it tends to learn the underlying geometric features which are able to be generalized across new objects and scenes.

Figure 12: Qualitative results of instance segmentation on ScanNet dataset. Although the model is trained on S3DIS dataset and then directly tested on ScanNet validation split, it is still able to predict high-quality instance labels.