[Reviews · NeurIPS 2019]

Reviewer 1



Authors propose a novel framework for instance segmentation on 3D point clouds. The idea is new and I haven’t seen it being used in 3D before. The idea is to predict a set of 3D bounding boxes from some global feature of the 3D scene, pair predicted bounding boxes with ground truth ones using optimal assignment algorithm and then calculate carefully designed loss. Quality: Generally, the quality of the work is good. The Computation Analysis section needs to be extended. Authors claim that their approach is approximately 10× more computationally efficient than (at least some) previous ones. Since the nature of approaches in 3D instance segmentation differs so much and different pre- and post-processing steps are used, it would be great to provide specific performance measurements for each step of the whole pipeline run at least for a couple of the solutions (e.g. with open code). E.g., many of the methods mentioned in this paper use Voxelization pre-processing, BlockMerging, Clustering, NMS, and true some of these can be costly, but on the other hand 3D-BoNet requires an additional run of SCN semantic segmentation - how does the cost of that compare to these other components? It would be much better to see some hard numbers on this comparison and not too hard to do (at least for the ones with code and are already prepped to run on Scannet). Clarity: The paper is well-written and structured clearly. The notation is good and overall paper is easy to read and understand. The code is going to be open-sourced, great! One of the key contributions of the paper is the ability to approximate the gradient for the Hungarian algorithm. This is a key component, and without an explicit statement of the gradient (more detailed than what is shown in the appendix) this method would be difficult to reproduce. Aside from that, it looks like it would be possible to reproduce paper results. Minor comments: Typo - “... Ai,j =1 iff the …” (page 4) SGPN [49] uses 1.5m×1.5m blocks for ScanNet training (see their C.1. Section). Originality: The paper presents an original and new method for 3D instance segmentation. The proposed approach correlates well with the proposal-free, per-point solutions for object detection that are gaining popularity in 2D. Significance: The results of the paper are valuable for people who research 3D instance segmentation problems (possibly 2D too). There aren’t many working approaches in this field and since 3D (2D) instance segmentation task is not as mature as say 2D object detection every new approach (especially an effective one) is certainly welcomed.

Reviewer 2



Overall, this paper proposes a novel and interesting idea, the writing is done well, and the numbers seem to back up most of the contributions. Below are my detailed comments. The proposed framework takes a 3D point cloud as input and outputs a fixed number of object hypotheses. For each hypothesis, the method regresses an axis-aligned bounding box, an objectness score, and a point mask. In order to set regression targets, the authors propose a novel association layer to associate each ground truth object to one object hypothesis. This is done by solving Hungarian matching on a hand-crafted pairwise cost matrix. To backprop through the non-differentiable matching process, the authors propose to estimate the gradient using Policy Gradient. As far as I know, incorporating Hungarian matching to associate hypotheses and ground truths as part of an end-to-end pipeline is novel. However, I feel the difference of this particular design was not emphasized enough neither in the writing and the experiments. For example, in terms of the one-to-one mapping, it makes sense to map every hypothesis to at most one ground truth object, but it is not as clear why we do not allow mapping one ground truth to multiple hypotheses. My concern comes from the fact that many well-known approaches (e.g. MaskRCNN) allow such ground truth “sharing”. I am aware that this might result in duplicated predictions and these methods often run NMS. Nonetheless, an empirical comparison one-to-one and one-to-many mapping would help justify the importance of this particular design choice. On a similar note, it would be interesting to see an empirical comparison between solving one-to-one mapping optimally and solving it greedily. I am also curious about what these MLPs learn. They might get mapped to different ground truth randomly after initialization. But after a while, do they learn to consistently respond to a specific part of the point cloud (or block)? My lack of understanding what these MLPs learn makes me wonder what happens if we apply the same idea presented in this paper to instance segmentation on images. Specifically, we would learn a model that takes an image and spits out scores, bounding boxes, and pixel masks. Though it seems viable algorithmically, I am not sure if it would work better than MaskRCNN. I wonder what the authors think about this idea. Regarding Algorithm 1: First, I was not sure how to interpret the step 1. My best guess is that it computes the squared distance of the n-th point to the min/max vertices of the i-th bounding box. If so, p_{xyz} should be a scalar value but why do we take the minimum over p_{xyz} in step 4? In addition, it looks like the probability is biased towards larger bounding boxes. I thought instead of Euclidean distance, it makes more sense to normalize distance w.r.t. each dimension.

Reviewer 3



3D Instance Segmentation is an interesting problem with a lot of possible applications. This paper proposes a single-stage, anchor-free and end-to-end trainable framework to address this problem. The bounding box association layer and the multi-criteria loss function are original designs, which might have a high impact on the research community. Unlike most of the instance segmentation works, which employ heavy post-processing steps, the proposed framework is able to give accurate predictions without any post-processing. The framework is novel and efficient, which might inspire more interesting works in this area. Besides the originality and significance, the paper is complete work, and the writing and illustration is clear and concise.

[Author Response · NeurIPS 2019]

**Paper Title**: Learning Object Bounding Boxes for 3D Instance Segmentation on Point Clouds

We would like to thank all reviewers for their very insightful comments and address them in the following.

**1. Comparison of computation efficiency.** Table 1 compares the time consumption of four existing approaches using
their released codes on the validation split (312 scenes) of ScanNet(v2) dataset. SGPN, ASIS, GSPN and our 3D-BoNet
are implemented by Tensorflow 1.4, 3D-SIS by Pytorch 0.4. All approaches are running on a single Titan X GPU and
the pre/post-processing steps on an i7 CPU core with a single thread. Note that 3D-SIS automatically uses CPU for
computing when some large scenes are unable to be processed by the single GPU. Overall, our approach is much more
    computationally efficient than existing methods, even achieving up to $20\times$ faster than ASIS.

Table 1: Time consumption of different approaches on the validation split (312 scenes) of ScanNet(v2) (seconds).

|  | SGPN | ASIS | GSPN | 3D-SIS | **3D-BoNet(Ours)** |
|---|---|---|---|---|---|
|  | network(GPU): 650 | network(GPU): 650 | network(GPU): 500 | voxelization, projection, | network(GPU): 650 |
|  | group merging(CPU): 46562 | mean shift(CPU): 53886 | point sampling(GPU): 2995 | network, etc. (GPU+CPU): | *SCN (GPU parallel): 208* |
|  | block merging(CPU): 2221 | block merging(CPU): 2221 | neighbour search(CPU): 468 | 38841 | block merging(CPU): 2221 |
| total | 49433 | 56757 | 3963 | 38841 | **2871** |

**2. Gradient estimation of Hungarian algorithm.** There are many ways to estimate the gradient of the bouding box
assignemnt. In our implementation we use a very simple approach and finding a better estimator is the scope of future
work. Given the predicted bounding box parameters as a stack vector of all the boxes, $\mathbf{B}$, and ground-truth boxes, $\bar{\mathbf{B}}$,
we compute the assignment cost matrix, $\mathbf{C}$. This matrix is converted to a permutation matrix, $\mathbf{A}$, using the Hungarian
algorithm. Here we focus on the euclidean distance component of the loss, $\mathbf{C}^{ed}$. The derivative of our loss component
w.r.t the network parameters, $\theta$, in matrix form is: $\frac{\partial \mathbf{C}^{ed}}{\partial \theta} = -2(\mathbf{AB} - \bar{\mathbf{B}}) \left[ \mathbf{A} + \frac{\partial \mathbf{A}}{\partial \mathbf{C}} \frac{\partial \mathbf{C}}{\partial \mathbf{B}} \mathbf{B} \right]^{T} \frac{\partial \mathbf{B}}{\partial \theta}$ (1) . The components
are easily computable except for $\frac{\partial \mathbf{A}}{\partial \mathbf{C}}$ which is the gradient of the permutation w.r.t the assignment cost matrix which is
zero nearly everywhere. We found that training the model works when setting this term to zero in our experiments.
However, convergence can be sped up using the straight-through-estimator [1], which assumes that the gradient of
the rounding is identity (or a small constant), $\frac{\partial \mathbf{A}}{\partial \mathbf{C}} = \mathbb{1}$. This speeds up convergence as it allows both the error in the
bounding box alignment (1st term of Eq. (1)) to be backpropagated and the assignment to be reinforced (2nd term of Eq.
(1)). This approach has been shown to work well in practice for many problems including for differentiating through
permutations for solving combinatorial optimization problems [4] and for training binary neural networks [2]. Other,
more complex approaches could also be used in our framework for computing the gradient of the assignment such as
[3] which uses a Plackett-Luce distribution over permutations and a reparameterized gradient estimator.

**3. One-to-one mapping vs. Many-to-one mapping.** The primary advantage of one-to-one mapping between predicted
boxes and ground truth is the computation efficiency during testing. Nevertheless, many-to-one mapping may bring
higher precision with sacrificing the speed. We agree that it is an interesting direction to integrate a greedy algorithm to
solve the one-to-one mapping problem, but it is non-trivial to make it differentiable.

**4. Discussion about what has been learnt.** Fundamentally, the designed multi-criteria loss functions for 3D bounding
box prediction enable the network to learn key vertices to include dense point clusters, thereby inferring an overall
boundary for the object. This general idea can indeed be extended to 2D images, as long as we are able to find good
metrics to measure the valid object pixels within a predicted box.

**5. Clarification of Algorithm 1.** w.r.t step 3 of Algo-
rithm 1, the probability $\boldsymbol{p}_{xyz} = \frac{1}{1+\exp(-\boldsymbol{\Delta}_{xyz})}$ is a vector,
e.g., $(p_x, p_y, p_z)$, indicating the probability of a point
being inside of the box from x-y-z axes. Eventually,
the minimum value of $(p_x, p_y, p_z)$ determines the final
probability of that point being inside of the box. This

Input PC    *Pred Mask #1*    *Pred Mask #2*    *Pred Mask #3*    *Pred Mask #4*    GT Masks

Figure 1: Visualization of predicted instance masks.

approximate probability is indeed biased towards slightly larger boxes, and we agree that a normalized distance is
worthwhile for future exploration and may benefit the bounding box prediction.

**6. Visualization of bounding boxes and point masks.** Figure 1 visualizes the predicted instance masks, where the
black points have $\sim 0$ probability and the brighter points have $\sim 1$ probability to be an instance within each predicted
mask. Predicted bounding boxes are visualized in the appendix (Section B) of our submission.

**References**

[1] Bengio, Y., Léonard, N., & Courville, A., *Estimating or propagating gradients through stochastic neurons .....* arXiv preprint arXiv:1308.3432 (2013).
[2] Yin, Penghang, et al., *Understanding Straight-Through Estimator in Training Activation Quantized Neural Nets*. ICLR (2019).
[3] Grover, Aditya, et al., *Stochastic Optimization of Sorting Networks via Continuous Relaxations*. ICLR (2019).
[4] Emami, P. and Ranka, S., *Learning permutations with sinkhorn policy gradient*. arXiv preprint arXiv:1805.07010 (2019).


[Meta-Review · NeurIPS 2019]

This paper presents an instance segmentation algorithm on 3D point cloud data, which predicts 3D bounding box per instance and membership per point. All reviewers agree to the novelty and significance of the proposed method. This is a good piece of work and deserves to be accepted as a spotlight.